# Defects in Cell Wall Differentiation of the Arabidopsis Mutant *rol1-2* Is Dependent on Cyclin-Dependent Kinase CDK8

**DOI:** 10.3390/cells10030685

**Published:** 2021-03-19

**Authors:** Isabel Schumacher, Tohnyui Ndinyanka Fabrice, Marie-Therese Abdou, Benjamin M. Kuhn, Aline Voxeur, Aline Herger, Stefan Roffler, Laurent Bigler, Thomas Wicker, Christoph Ringli

**Affiliations:** 1Department of Plant and Microbial Biology, University of Zurich, 8008 Zurich, Switzerland; isabel.schumacher@botinst.uzh.ch (I.S.); nf.tohnyui@unibas.ch (T.N.F.); MarieTherese.Abdou@kssg.ch (M.-T.A.); benkuhn@gmx.de (B.M.K.); alinegalatea.herger@botinst.uzh.ch (A.H.); stefan.roffler@botinst.uzh.ch (S.R.); wicker@botinst.uzh.ch (T.W.); 2Zurich-Basel Plant Science Center, 8092 Zurich, Switzerland; 3Institut Jean-Pierre Bourgin, INRAE, AgroParis Tech, Université Paris-Saclay, 78000 Versailles, France; Aline.Voxeur@inrae.fr; 4Department of Chemistry, University of Zurich, 8057 Zurich, Switzerland; laurent.bigler@chem.uzh.ch

**Keywords:** cell wall, cell wall differentiation, cell corner, ultrastructure, callose, *rol1-2*, *RHM1*, pectin, *CDK8*, *surr*, *suppressor or rol1-2 root developmental defect*

## Abstract

Plant cells are encapsulated by cell walls whose properties largely determine cell growth. We have previously identified the *rol1-2* mutant, which shows defects in seedling root and shoot development. *rol1-2* is affected in the *Rhamnose synthase 1* (*RHM1*) and shows alterations in the structures of Rhamnogalacturonan I (RG I) and RG II, two rhamnose-containing pectins. The data presented here shows that root tissue of the *rol1-2* mutant fails to properly differentiate the cell wall in cell corners and accumulates excessive amounts of callose, both of which likely alter the physical properties of cells. A *surr* (*suppressor of the rol1-2 root developmental defect*) mutant was identified that alleviates the cell growth defects in *rol1-2*. The cell wall differentiation defect is re-established in the *rol1-2 surr* mutant and callose accumulation is reduced compared to *rol1-2*. The *surr* mutation is an allele of the *cyclin-dependent kinase 8* (*CDK8*), which encodes a component of the mediator complex that influences processes central to plant growth and development. Together, the identification of the *surr* mutant suggests that changes in cell wall composition and turnover in the *rol1-2* mutant have a significant impact on cell growth and reveals a function of CDK8 in cell wall architecture and composition.

## 1. Introduction

Growth of plants and individual cells require the coordination of numerous processes taking place in the symplast and apoplast alike. Plant cells are surrounded by cell walls, which are rigid yet flexible structures that provide protection and determine the size and shape of the cell. Consequently, formation, expansion, and constant remodeling of the cell wall is an intrinsic part of plant cell developmental processes. Primary cell walls are predominantly composed of the polysaccharides cellulose, hemicellulose, and pectins, and structural proteins, which all influence the physical properties. The exact composition can vary considerably among cell and tissue types depending on the specific requirements or developmental stage [1,2,3].

Pectins have gel-like properties and can embed other cell wall components. A number of interactions of pectins with other cell wall components have been identified that influence the mechanical and physical properties of the cell wall. The different functions of pectins can, in part, be explained by diverse subgroups among pectins that show distinct compositions. The pectin homogalacturonan (HG) is a galacturonic acid polymer, which is produced and deposited in methylesterified form. Demethylesterification results in negatively charged carboxyl groups that can subsequently form ionic bonds with Ca2+ to stabilize the structure, but at the same time has been reported to be the starting point of degradation of the HG polymer, enabling remodeling of the pectin fraction. Rhamnogalacturonan I (RG I) has a backbone of α 1,4-linked Gal-Rha dimers with sidechains composed of either Ara or Gal. RG I is thought to loosely interact with other cell wall components and rather influence the porosity of the cell wall, which is a determinant of the permeability for wall-modifying enzymes. RG II, on the other hand, has a α 1,4-linked Gal backbone, has structurally complex side chains composed of a number of different sugars. It influences the mechanical properties of cell walls through ionic interactions with boron which help to crosslink RG II [4,5].

The role of pectin structures in determining architecture and physical properties of cell walls is frequently investigated by analyzing mutants that are affected in one or several pectin structures. Altering the supply of individual sugar components, modifying the glycosyltransferases producing the different sugar linkages or pectin-modifying enzymes such as pectin methylesterases and their inhibitory proteins [6,7,8] are different possible strategies. In our previous work, we identified *rol1* (*repressor of lrx1*) mutants based on their ability to alleviate the root hair formation defect induced by mutations in LRR-extensin 1 [9,10]. LRXs are extracellular receptors of Rapid ALkalinization Factor (RALF) peptide and signaling partner of the transmembrane receptor kinase FERONIA and involved in cell wall integrity signaling [11,12,13,14]. *ROL1* codes for the Rhamnose synthase 1, RHM1, and *rol1* mutants produce aberrant pectin structures with reduced labelling of the RG I-specific antibody LM5 and a reduced level in the RG II-specific sugars O-methyl-fucose and O-methyl-xylose [10]. FER has been shown to bind pectin [15], and changes in pectin structures are signaled via the LRX/FER pathway [16]. Altering the pectin structures due to the *rol1* mutations might influence the LRX/FER signaling process which results in the suppression of *lrx1*. *rol1-2* is a missense allele of *ROL1* (At1g78570) and induces a considerable alteration in cell growth and development compared to the wild type, with shorter roots and root hairs, epinastic cotyledons, and brick-shaped rather than jigsaw puzzle-like cell shapes in epidermal cells on the adaxial side of cotyledons [17]. In addition to the changes in pectin, the *rol1-2*-induced defect in Rha synthesis also influences the accumulation of flavonols, secondary metabolites derived from phenylpropanoids that are glycosylated mainly by Glc and Rha [18]. Flavonols are known to modulate cell growth processes both in plants and animals, and, in plants, influence the transport activities of auxin, an abundant plant hormone that influences many different processes including cell growth [19]. The altered flavonol glycosylation profile in *rol1-2* mutants has been revealed to be a main cause of the growth defects observed in *rol1-2* shoots. Blocking flavonol biosynthesis in *rol1-2* by mutating genes coding for enzymes of the biosynthesis pathway such as Flavonol synthase 1 and Chalcone synthase (FLS1 and TT4, respectively) suppresses the aberrant shoot development [17,20]. By contrast, the defect in root development was largely unaffected in these lines, suggesting that these growth defects are flavonol-independent and predominantly caused by alterations in the cell walls.

This work demonstrates that the cell wall structure is modified in *rol1-2* mutant roots, which are visible as ultrastructural alterations as well as ectopically accumulating callose. These changes are likely responsible for the reduced elongation growth in *rol1-2* root tissue. In a forward genetic screen, a *surr* (*suppressor of the rol1-2 root developmental defect*) mutant was identified that alleviates the growth defects of *rol1-2* and re-establishes the cell wall structure and differentiation in the root. The *surr* mutation was identified as a new allele of CDK8/CDKE1 (At5g63610), a cyclin-dependent kinase that is part of the mediator complex [21] and influences fundamental developmental processes. Hence, CDK8 and, consequently, the mediator complex are involved in the regulation of cell wall differentiation processes which ultimately influence cell and tissue growth properties.

## 2. Materials and Methods

### 2.1. Plant Growth and Mutagenesis

Seeds of the Arabidopsis *rol1-2* mutant were mutagenized with ethyl methanesulfonate (EMS) and propagated for M2 generation as described [20]. M2 seedlings were grown on half-strength MS plates, containing 2% sucrose, myo-inositol, and vitamins [22], 0.6% Phytagel (Sigma, Buchs, Switzerland) in a vertical orientation for seven days at 22 °C, 16 h light and 8 h dark. Seedlings developing longer roots than the *rol1-2* control were selected. They were transferred to soil and grown under the same light and temperature regime for propagation and crossing.

The different mutant lines are described in [10] for *rol1-2*, [23] for *rao1-1* and *rao1-2*, and [24] for *cdk8-1*. The primer pairs used for PCR-based molecular markers for the different mutations are listed (Appendix A).

### 2.2. Microscopic Analysis of Plant Growth Phenotypes

To quantify root length, plants were grown in a vertical orientation on a medium described above for the number of days indicated in the corresponding figure legends; the plates were scanned and root length was measured using ImageJ. For root hair length determination, seedlings were grown the same way as for root length measurements, pictures of root hairs were taken with an MZ125 Binocular (Leica; Heerbrugg, Switzerland) using a DFC420 digital camera (Leica), and root hairs in the focal plane were used for measurements using ImageJ. For epidermal cell length analysis, six seedlings of each line were observed under a Zeiss Axio Imager Z1 microscope (Zeiss, Feldbach, Switzerland) and pictures of root cells were taken from the differentiation zone to ascertain completed cell elongation. The length of five or more epidermal cells per seedling was measured using ImageJ.

### 2.3. Quantitative RT-PCR

For gene expression analysis, total seedlings of the different genotypes were grown in a vertical orientation as described above and used for extraction of total RNA using the SV total RNA isolation kit (Promega, Dübendorf, Switzerland). This total RNA (300 ng per sample) was used for RT-PCR using the iScript advanced cDNA kit (BioRad). Quantitative RT-PCR was performed on a CFX96TM real-time system (Bio-Rad) with the Kapa Syber Fast qPCR (Kapa Biosystems, Basel, Switzerland) technology. *EFα*, *GAPDH*, and *UBI10* were used as internal standards to quantify expression. Data analysis was carried out with CFX Manager 3.1 software (Bio-Rad, Cressier, Switzerland).

### 2.4. Aniline-Blue Staining for Callose Detection in Whole Seedlings

Roots were fixed in PEM buffer (4% paraformaldehyde in 1 M NaOH, 50 mM PIPES, 1 mM EGTA and 5 mM MgSO4), then rinsed three times with 100 mM Na-phosphate buffer (pH 8). The tissues were stained directly before microscopy with 0.1% methyl blue (certified for use as aniline blue; Sigma) in 100 mM Na-phosphate buffer. Images were acquired using a Leica DM 6000 epifluorescence microscope equipped with an Andor Neo 5.5 sCMOS camera (Andor Technology Ltd., Belfast, UK).

### 2.5. Ultrastructural Analysis and Immunogold Labelling

Roots were fixed overnight in a solution of 3% formaldehyde and 1.25% glutaraldehyde in 0.05% cacodylate buffer, postfixed in 2% OsO4 for two hours. Serial dehydration was carried out in increasing concentrations (for 10 min each, 30%, 50%, 70%, 90% and 2× in 100% *v/v*) of acetone in water, the roots were infiltrated overnight with 50% *v/v* Epon/acetone, and then embedded in 100% Epon resin. A very detailed step-by-step description has been previously published [25]. For immunolabelling, ultrathin sections produced of material embedded as described above were incubated overnight with 1:150 dilution of the anti-(1,3)-β-glucan antibody against callose (Biosupplies, Bundoora, Australia) in 4% nonfatted milk in PBS buffer (pH 7.2). Then, they were rinsed and labelled for one hour in 1:25 dilution of the antimouse secondary antibody conjugated to 10 nm gold particles in 4% nonfatted milk in PBS buffer (pH 7.2). The sections were poststained with 1% uranyl acetate for 15 min and 1% lead citrate for 10 min prior to visualization in the TEM (FEI CM100, Amolf, Amsterdam, The Netherlands) using a Gatan Orius 1000 CCD camera (Amolf, Amsterdam, The Netherlands) using a Gatan Orius 1000 CCD camera (Gatan, Munich, Germany).

### 2.6. Plant Infection Experiments

The infection experiments with *Botrytis cinerea* were performed as described [26]. In brief, the fungus was grown on potato dextrose agar at 23 °C under continuous light. After 10 days, a dense carpet of conidia was formed, which were adjusted to a final concentration of 3.10^5^ conidia/mL, of which twenty microliter drops were placed on Arabidopsis leaves of 5-week-old plants. Leaf areas were measured by ImageJ and statistical analysis was carried out by means of a Kruskal-Wallis test using GraphPad Prism.

### 2.7. Flavonol Extraction and Quantification

Extraction and quantification of flavonols from Arabidopsis seedling shoots was performed as described in detail in [20]. In brief, shoots of one hundred 6-days-old seedlings were lyophilized to determine the dry weight and the material was extracted with 80% methanol. High-performance liquid chromatography electrospray ionization mass spectrometry (HPLC-ESI-MS) experiments were performed on an Acquity UPLC (Waters, Milford, MA, USA) connected to a Bruker maXis high-resolution quadrupole time-of-flight mass spectrometer (Bruker Daltonics, Bremen, Germany). An Acquity BEH C18 HPLC column (1.7 µm, 2.1 × 100 mm fitted with a 2 × 32 mm guard column) was used with a gradient of solvent A (water, 0.1% (*v/v*) HCOOH) and solvent B (CH3CN, 0.1% (*v/v*) HCOOH; 0.45 mL flow rate, linear gradient from 5% to 95% B within 30 min). The area under each flavonol peak (identified by the mass and missing peak in the *rol1-2 fls1-3* sample) was used for quantification.

## 3. Results

### 3.1. Cell Wall Defects in the rol1-2 Mutant Alter Root Growth

An obvious characteristic feature of the *rol1-2* mutant is the reduced elongation of both root hairs (Figure 1a,b) and roots (Figure 1c), confirming our previous findings [10]. The reduced root length can be explained by the reduced elongation growth as exemplified by the cell length of trichoblasts, root hair-forming root epidermal cells. Compared to the wild type, *rol1-2* trichoblasts are significantly shorter (Figure 1d), explaining the apparent increase in root hair density seen in Figure 1a.

### 3.2. The rol1-2 Mutation Causes Modifications of the Cell Wall Composition and Structure

By analyzing root cell wall structures, we aimed to investigate possible changes in cell wall development due to the *rol1-2* mutation. A common stress response upon alterations in cell wall composition is the production of callose to reinforce the cell wall [27,28]. Visualization of callose in the root by aniline blue revealed significant amounts of callose in the *rol1-2* mutant which was not observed in the wild type (Appendix A). Hence, the *rol1-2* mutant appears to deposit callose, in the cell walls, possibly as a strategy to reinforce these structures.

The ultrastructure of root cell walls was analyzed in more detail. Due to the observed defect of *rol1-2* mutants in root cell elongation (Figure 1d), we wanted to investigate possible alterations in both the meristematic and the differentiation zone of the root. In the meristematic zone of both wild-type and *rol1-2* mutants, the cell wall and intercellular space are regularly structured (Figure 2a,b). In this tissue, cells formed and did not undergo major growth processes. In the differentiation zone, by contrast, cell walls of the wild type underwent major rearrangements, restricting as well as delineating the dense and regular structure to the cell wall between cells. The intercellular space of cell corners changed notably at the ultrastructural level, appearing less electron-dense (Figure 2d). In the *rol1-2* mutant, however, the intercellular space remained densely packed, suggesting incomplete differentiation (Figure 2e).

Callose depositions in cell walls were identified using a callose-binding module (CBM; for details, see Material and Methods). In contrast to the wild type where no significant antibody labelling was detected, the *rol1-2* mutant showed generally increased binding of the antibody in the cell wall with intermittent occurrence of local callose depositions instead of proper incorporation into the cell walls (Figure 3a). Quantification of gold particles confirmed differences among the different genetic backgrounds. The comparison of cell walls between neighboring cells versus those in cell corners also showed that the difference in callose deposition was found for all cell walls but total amount of callose is higher in cell corners (Figure 3a, note the interruption of the scale on the y-axis.).

Due to the reduction in root hair elongation in the *rol1-2* mutant, the ultrastructure of this particular cell type was also analyzed. Root hair cell walls of the wild type are compact and have a more electron-dense appearance than those of the *rol1-2* mutant—they are thicker but less electron-dense (Figure 3b). Cell wall material of *rol1-2* root hairs was decorated with the anticallose CBM (Figure 3b), which was not observed in the wild type. This labelling confirms the aniline staining of whole seedling roots which indicates increased callose depositions in different root tissues (Appendix A). Hence, callose appears to be deposited into the cell walls as a general response to alterations in cell wall composition and/or architecture induced by *rol1-2*.

### 3.3. Alleviation of the rol1-2 Mutant Phenotype

To identify genes/proteins that are involved in the establishment of compensatory changes in *rol1-2* mutant cell walls, a genetic screen was conducted. *rol1-2* seeds were mutagenized with ethyl methanesulfonate and propagated to the M2 generation as previously described [20]. To identify modulators of the cell wall-related phenotype of the *rol1-2* mutant, plants with an altered *rol1-2* root (hair) phenotype were isolated. The short roots and short root hairs of *rol1-2* compared to the wild type (Figure 1a,b) are distinct characteristics, allowing one to readily identify individuals that develop alleviations of these *rol1-2* phenotypes. In this mutant screen, one line was identified that showed clear suppression of the root (hair) developmental defects of *rol1-2* and was thus named *surr* (*suppressor of the rol1-2 root developmental defect*). The identified line was backcrossed twice with *rol1-2* before being analyzed in more detail. In a segregating F2 population obtained from a backcross with *rol1-2*, around one-quarter of the progenies showed a *rol1-2 surr* phenotype indicating that *surr* is a recessive mutation. *rol1-2 surr* was also crossed with a Col wild-type plant to obtain *surr* single mutants. Seedlings of the *rol1-2 surr* line showed an increase in root and root hair length compared to *rol1-2*. The short root hair phenotype of *rol1-2* was fully suppressed, with *rol1-2 surr* and the wild type having comparable root hair lengths (Figure 1a,b). By contrast, the root length of *rol1-2 surr* is intermediate—it is significantly longer than *rol1-2* but shorter than the wild type (Figure 1c). The reduced epidermal cell length of *rol1-2* trichoblasts was also alleviated with *rol1-2 surr* developing trichoblasts of comparable lengths as the wild type (Figure 1d), which can also be seen in the root where root hairs are further apart than in *rol1-2* (Figure 1a).

The *surr* mutation not only alleviates the altered root growth in *rol1-2* but also has an effect on shoot development. Wild-type seedlings develop epinastic cotyledons and epidermal pavement cells that form jigsaw puzzle-like cell shapes (Figure 4a). *rol1-2* cotyledons, by contrast, are hyponastic and pavement cells that have lost the typical lobing (Figure 4b; [17]). *rol1-2 surr* double mutant seedlings develop epinastic cotyledons and hence have a suppressed *rol1-2* phenotype, but in the vast majority brick-shaped pavement cells as the *rol1-2* (Figure 4c). Hence, while cotyledon formation of *rol1-2* is effectively suppressed by *surr*, the cell shape phenotype is not.

In the next step, it was investigated as to whether the *surr* mutation has an effect on cell wall development of *rol1-2*. To this end, aniline-blue staining was conducted on seedling roots. In contrast to the strongly staining *rol1-2* mutant, aniline blue staining in *rol1-2 surr* double mutants was only found interspersed and not in a regular pattern (Appendix A).

The ultrastructural analysis revealed that in the differentiation zone of the root, *rol1-2 surr* seedlings underwent rearrangement of the cell wall in cell corners (Figure 2). Hence, the differentiation of cell corners between the meristematic and the differentiation zones missing in *rol1-2* was re-established in *rol1-2 surr* seedlings (Figure 2f). This observation is paralleled by reduced labelling of cell walls with the anticallose CBM (Figure 3) and the absence of callose deposits which were observed in the *rol1-2* mutant. Finally, the aberrant cell wall formation in *rol1-2* root hairs compared to the wild type was found to be largely alleviated, with *rol1-2 surr* root hairs having a compact and thin appearance akin to the wild type in addition to very infrequent labelling with anticallose CBM. It would have been interesting to investigate labelling of cell corners with the LM5 antibody binding to Gal-side chains of RGI since LM5 labelling was previously shown to be altered in the *rol1* mutant compared to the wild type [10]. However, binding of the LM5 antibody failed in all plant lines, probably due to embedding technology used here. Together, these data revealed that the defects in cell wall structures found in *rol1-2* are largely suppressed in *rol1-2 surr*.

### 3.4. The Surr Mutation Is a New Allele of the Cyclin-Dependent Kinase 8

To identify the *surr* mutation, a whole-genome sequencing approach was chosen. To this end, the *rol1-2 surr* mutant was backcrossed with *rol1-2*, propagated to the F2, among which 12 individuals showing the *rol1-2 surr* mutant phenotype were selected and propagated to F3. These 12 F3 families were confirmed to be homozygous for the *rol1-2 surr* phenotype, and plants of the different families were pooled for DNA extraction and whole-genomic sequencing. The recessive nature of the *surr* mutation (see above) allowed us to conclude that all the selected F3 families should be homozygous for the *surr* mutation. The sequencing data were aligned with the wild-type Columbia sequence available at TAIR (www.arabidopsis.org) and revealed few single-nucleotide polymorphisms (SNPs) in the *rol1-2 surr* mutant, one being the *rol1-2* mutation, and four mutations at the lower end of chromosome 5—namely, in *nucleotide di-kinase 2* (*NDPK2*), *cyclin-dependent kinase 8* (*CDK8*), *alpha-mannosidase 2* (*MAN2*), and *TPR* a member of the *tetratricopeptide*-like superfamily (Figure 5a). The cosegregation of several genetically linked mutations indicated that one of these SNPs might induce the *surr* phenotype. To genetically separate the different SNPs, among the F2 population segregating for the *surr* phenotype, an additional 80 seedlings showing the *rol1-2 surr* phenotype were selected; their DNA was extracted, and the SNP in *CDK8* was the only one that was found to be homozygous in all F2 individuals. For all other SNPs, plants with recombinant chromosomes were identified—i.e., plants being heterozygous for one or more of the other SNPs while being homozygous for the *surr* mutation. The cosegregation of the mutation in *CDK8* with the *surr* phenotype led us conclude that *surr* is a new allele of *CDK8*. The *surr* mutation represents a missense allele that changes the glycine at position 141 to a glutamic acid, near the kinase active site and the residue affected in the *rao1-1* allele (Figure 5b). Alignment of CDK8 homologs of different plant species and humans shows that this Gly residue is completely conserved across very different species (Appendix A).

To confirm that a mutation in *CDK8* can suppress *rol1-2* phenotypes, *rol1-2* was crossed with the two *CDK8* alleles, *rao1-1* and *rao1-2,* that were previously isolated as modifiers of mitochondrial retrograde signaling [23]. The positions affected in these two missense alleles are indicated in Figure 5b, with *rao1-1* being affected in the well-conserved kinase active site close to *surr* (Appendix A). As single mutants, all three *cdk8* alleles showed reduced root growth compared to the wild type and *rao1-1* and *rao1-2* alleviated the short root phenotype of *rol1-2* (Figure 6a) comparable to *surr* (Figure 1c). A clear difference between *surr* and the two *rao1* alleles was observed in root hair development, where root hair length in *surr* is comparable to the wild type whereas both *rao1* alleles develop very short root hairs, suggesting that the *rao1* mutations have a stronger impact on CDK8 activity than *surr*. Both *rao1* alleles, however, alleviate the *rol1-2* growth defects, resulting in longer root hairs than either *rol1-2*, *rao1-1*, or *rao1-2* (Figure 6b). Additionally, the hyponastic cotyledon phenotype of *rol1-2* is suppressed by the *rao1* alleles since both *rol1-2 rao1* double mutants form epinastic cotyledons (Figure 6c). The comparable effect of *surr* and the two independent *rao1* alleles led us conclude that the *surr* mutation in *CDK8* induces the observed suppression of *rol1-2*.

CDK8 was also shown to be involved in the resistance response to *Botrytis cinerea* [24] and we expected that the *surr* mutation might have a similar effect. Hence, wild-type Col, *rol1-2*, and *rol1-2 surr* mutants were tested for susceptibility to *B. cinerea*. As controls, we also included *cdk8-1*, the T-DNA allele of CDK8, and *cdk8-1* complemented with a *35S:CDK8myc* construct [24]. Col and *rol1-2* showed a similar infection rate, suggesting that *rol1-2* does not influence the defense response (Figure 7a). The *surr* mutation increased the resistance in the *rol1-2 surr* line, indicating that this *cdk8* allele has a similar effect as the T-DNA insertion allele *cdk8-1*. The positive control with the complemented *cdk8-1* line again showed the expected increased susceptibility [24]. The comparable effects of *surr* and *cdk8-1* with respect to the resistance response to *B. cinerea* infection again suggests that *surr* is allelic to *cdk8*.

### 3.5. Surr Leaves RHM2 or RHM3 Expression and Accumulation of Flavonols Unaffected

CDK8 has been shown to influence gene expression [23,24], which can modify physiological activities including the biosynthesis of compounds that are relevant for the development of the *rol1-2* mutant phenotype. To investigate whether the *rol1-2* mutation, which is in the rhamnose synthase *RHM1*, is compensated by increased expression of the *RHM1* homologs *RHM2* and *RHM3*, quantitative RT-PCR was performed on seedlings of the different lines. This analysis, however, did not reveal a change in gene expression in *rol1-2* or *rol1-2 surr* compared to the wild type that would indicate a compensatory upregulation in the expression level of the *RHM1* homologs as the basis of suppression of *rol1-2* by *surr* (Figure 7b).

Alternatively, the suppression of the *rol1-2* shoot phenotype by *surr* might be due to a strong reduction in flavonol biosynthesis. Mutations in *FLS1*, the major flavonol synthase of Arabidopsis, results in a complete suppression of the *rol1-2* shoot phenotype while the *rol1-2* root phenotype is not modified [20]. Quantification of the flavonol content in seedling shoot tissue revealed a reduced total flavonol content in *rol1-2* compared to the wild type, confirming previous findings [29] and an almost flavonol-less *rol1-2 fls1-3* double mutant. The *rol1-2 surr* double mutant shows some reduction in flavonol content compared to *rol1-2* of about 25% (Figure 7c), which is not comparable to the reduction observed in *rol1-2 fls1-3* and thus highly unlikely the cause of the partial suppression of the *rol1-2* shoot phenotype in *rol1-2 surr* mutant lines.

## 4. Discussion

The root represents a developmental gradient of cells, with undifferentiated cells at the root apical meristem, followed by elongating cells, and subsequently, after completed elongation, differentiation into the mature cell types [30]. It is well-established that, during this developmental process, cell walls are constantly modified to adjust to growth direction, increased cell elongation, and changing requirements of physical stability to resist internal turgor pressure emanating from the cell they encapsulate [3]. This system coordinating cell wall modifications and cell growth seems out of balance in the *rol1-2* mutant which develops short roots and root hairs (Figure 1).

During elongation growth, the pectin matrix is modified and the cellulose microfibrils are oriented transversely relative to the growth axis [31,32]. Expansins are induced which allow nonenzymatic enlargement of the cell wall [33,34] and Xyloglucan-modifying enzymes accumulate to modify the cellulose–hemicellulose network [35,36]. Processes requiring the locally restricted weakening of cell walls are paralleled by pectinase activity that degrade the pectin matrix [37,38] and, in roots, are observed during lateral root formation [39]. Similarly, the modification of the mechanical properties of cell walls can be an initiating signal for the formation of organs in the shoot, emphasizing the importance of mechanical signals for plant development [40,41].

### 4.1. Lack of Cell Wall Differentiation in rol1-2 Roots

Cell wall-modifying and -degrading processes are likely to also lead to the observed modification of cell wall material in cell corners of wild-type roots which are compact in the undifferentiated meristematic zone but much less dense in the differentiation zone. This process is impaired in the *rol1-2* mutant where cell corners appear to remain undifferentiated (Figure 2). The specific labelling of cell wall epitopes in cell corners by antibodies raised against different cell wall components in different plants suggests that these structures have specific functions requiring particular cell wall architectures. For example, the antibodies LM7, detecting demethylesterified pectins, and PAM1, detecting galacturonans, label cell corners. Several structural cell wall proteins, including proline-rich and glycine-rich structural proteins, are specifically deposited in cell corners, suggesting that they are important for the mechanical properties of the tissue [42,43,44,45]. Intercellular spaces are produced to allow proper elongation growth and a reduction in cell separation forces [46]. The *rol1-2* mutant fails to turnover cell wall material in cell corners which usually takes place in the wild type both in root (shown here) and the shoot [47], resulting in persistently dense wall material in the *rol1-2* mutant. Most likely, this failure of developmental modification influences cell wall stability. As a consequence, the cell walls in *rol1-2* roots do not yield to the turgor pressure, resulting in reduced elongation growth. A more detailed analysis of the changes in cell wall composition induced by *rol1-2* was previously performed and revealed a reduction in labelling with the antibody LM5 that recognizes Gal sidechains of RGI, and a reduction in the RGII-specific sugars O-methly-fucose and O-methyl-xylose [10]. In petals, the *rol1-2* mutant shows reduced Rha levels in cell wall fractions [48], again indicating an effect on structure and/or abundance of RGI and RGII. These alterations in Rha-containing polymers are in agreement with *rol1-2* presenting an allele of *RHM1*, one of three Rha-synthases encoded by the Arabidopsis genome [49].

A function of RGI sidechains is to regulate cell wall porosity and thus permeability for cell wall-modifying enzymes, which is of particular importance during cell expansion [50]. RGI production is prominent in the elongation zone, and delivery and orientation in root cell walls is developmentally regulated [51]. Hence, the lack of cell wall modification in *rol1-2* root cell corners possibly is a consequence of the altered RGI which would explain the reduced cell growth observed in the root of *rol1-2* [10,48].

It is not clear whether the increased callose deposition in the *rol1-2* root (Figure 2,Figure 4) is a consequence of the failure to modify the cell wall, but it certainly signifies a deviation in cell wall development. The induction of callose deposition is a well-known compensatory reaction of cells to defects in cell wall development [27,28]. In the *rol1-2* context, the deposition of callose might further reduce the permeability of the cell wall for enzymes, and thus inhibit the differentiation process and elongation growth. The increased callose deposition in *rol1-2* does not positively influence the response to infection with *B. cinerea* (Figure 7). Callose deposition at sites of pathogen attack is a well-described phenomenon [52]. Yet, a correlation of callose deposition and resistance is not always observed [53]. Similarly, *rol1-2* overaccumulates callose, yet is as susceptible to *B. cinerea* infection as the wild type. Mutations in *cdk8*, on the other hand, increase resistance, also when reducing the callose content in the context of the *rol1-2* mutant. In fact, *cdk8* was shown to increase the resistance against *B. cinerea* by influencing the development of the cutin layer as a barrier against pathogen invasion [24].

### 4.2. CDK8 Is Involved in Establishing the Aberrant Development of rol1-2

The characterization of the identified *surr* mutant revealed that it largely reverts the lack of wall modification in root cell corners and in root hairs and, thus, alleviates the growth defects of *rol1-2* (Figure 2,Figure 3). Additionally, the *rol1-2* developmental defect in cotyledons is partly suppressed, resulting in wild type-like epinastic cotyledons (Figure 4). The pavement cells, however, still form brick-like cell shapes (Figure 4), which is typical for the *rol1-2* mutant and mainly influenced by flavonols [17,20]. The *rol1-2* mutant induces altered accumulation of flavonols and blocking of this group of metabolites fully suppresses the *rol1-2* shoot phenotype without changing the defects in root development [17,20]. Hence, the effect of *surr* is mainly on cell wall differentiation, a view that is supported by the very modest effect of *surr* on flavonol accumulation (Figure 7).

The identified *surr* mutant shows effects on plant development and plant–pathogen interaction similar to other independently identified alleles of the *cyclin-dependent kinase 8* (*CDK8*, also known as *CDKE1*) (Figure 7) [23,24], indicating that the *cdk8* mutation in the *surr* mutant is causing the observed suppression of *rol1-2*. *rao1-1* and *rao1-2* develop very short root hairs, which is a distinct phenotype of these alleles not observed in the *surr* mutant. This suggest that the *rao1* alleles have a stronger effect than *surr*. The amino acid substitutions induced by *rao1-1* and *rao1-2* are in the kinase active site and the ATP binding site, respectively [23]. While *surr* is located near *rao1-1*, it appears that the kinase active site is not directly affected (Appendix A). If the amino acid substitutions in the *rao1* alleles interfere with protein activity yet maintain protein stability, this can interfere with the protein–protein interaction network involving CDK8 [54,55] and cause the observed strong effect on root hair development.

CDK8 has been implicated in a number of different processes including floral organ determination, mitochondrial retrograde signaling, JA signaling and plant–pathogen interactions [23,24,56,57]. It is a central component of the mediator complex that is evolutionarily conserved among eukaryotic species and links transcription factors with RNA Pol II [21], and, in plants, was recently shown to regulate ABA-related gene expression [55]. It seems likely that mutations in *cdk8* modify gene expression in the *rol1-2* mutant, resulting in suppression of several of the *rol1-2* phenotypes, possibly including genes that are misregulated in the *rol1-2* mutant compared to the wild type. The effect of mutations in *rol1* or *cdk8* on gene expression has been studied by different groups and revealed thousands of affected genes including genes involved in cell wall formation, modification and the sugar interconversion pathway for the synthesis of the substrates for the production of cell wall polysaccharides. These include a range of different proteins including but not limited to enzymes involved in pectin biosynthesis, -modification, and -turnover, such as pectin esterases, pectin methyltransferase inhibitors, pectin lyases, RG xylosyltransferases and other glycosyltransferases [6,10,23,24]. The identification of the gene(s) among this large collection relevant for suppression of *rol1-2* is beyond the frame of this project; yet, two obvious candidate genes were investigated—namely, the *ROL1*/*RHM1* homologs *RHM2* and *RHM3* [49], the overexpression of which might compensate for the absence of *RHM1* function. However, since *RHM2* and *RHM3* expression is not increased beyond wild-type levels in the *rol1-2* and *rol1-2 surr* mutant lines (Figure 7b), this postulated compensatory mechanism appears to not be responsible for the observed effect of the *surr* mutation.

## 5. Conclusions

In summary, we have identified a cell wall differentiation defect in *rol1-2* root tissue that plausibly causes the observed *rol1-2* mutant phenotypes. The failure of *rol1-2* plants to drastically modify wall material in cell corners likely has an effect on physical properties of this tissue and, as a consequence, on cell elongation. This hypothesis is supported by the identification of a mutation in *cyclin-dependent kinase 8* (*CDK8*) that suppresses the cell wall differentiation defect and the root growth phenotype of the *rol1-2* mutant. As part of the multiprotein mediator complex regulating transcriptional activity, CDK8 influences the expression of cell wall-related genes that modify cell wall differentiation processes important for the growth and development of individual cells, tissues, and entire plants.

## Figures and Tables

**Figure 1 cells-10-00685-f001:**
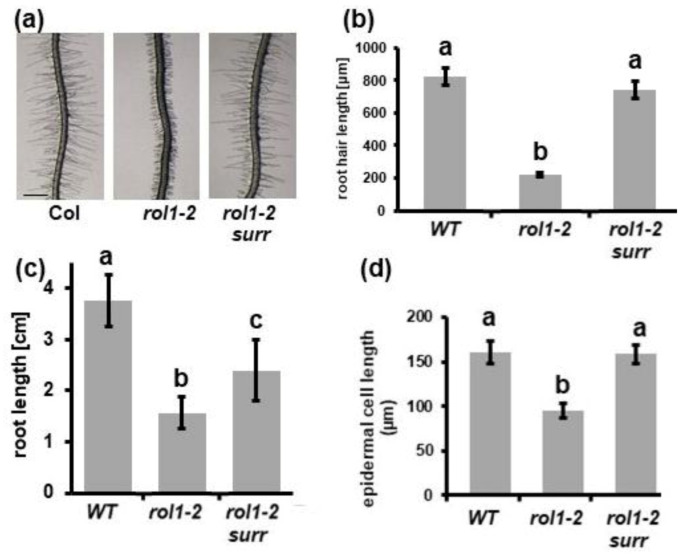
The reduced elongation of roots and root hairs of *rol1-2* are suppressed by the *surr* mutation. Seedlings were grown for six days in a vertical orientation. (**a**) Wild-type Columbia (Col) seedlings developed roots with long root hairs, whereas the *rol1-2* mutant developed shorter root hairs. This phenotype was suppressed in the *rol1-2 surr* mutant. (**b**) Quantification of root hair length of the different lines shown in panel (**a**). (**c**) A comparable effect was observed in the main root that is longer in wild-type Col than *rol1-2*, while *rol1-2 surr* shows intermediate root growth. (**d**) Cell length of trichoblast epidermal cells is shorter in *rol1-2,* whereas Col and *rol1-2 surr* are not significantly different. The different letters above the columns indicate significant differences (student *t*-test, *p* < 0.05, *n* > 10). Error bars represent standard deviations.

**Figure 2 cells-10-00685-f002:**
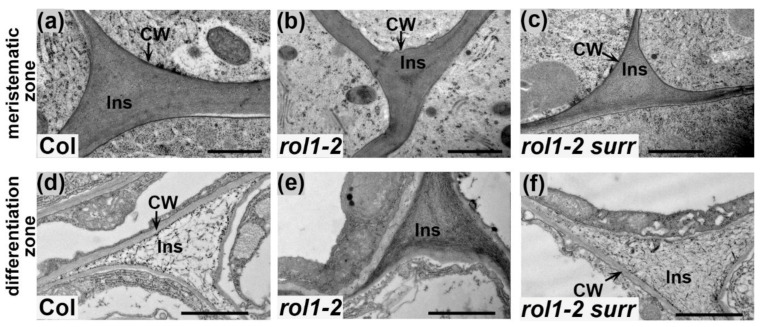
Changes in cell wall architecture in root tissue. Electron microscopic analysis of the cell wall ultrastructure in seedling root tissue. The wild-type cell wall architecture is modified between the meristematic tissue with electron-dense cell walls (**a**) and differentiation zone with much less dense material (**d**). This differentiation is absent in *rol1-2* (**b**,**e**), and re-established by the *surr* mutation (**c**,**f**). CW: cell wall; InS: intercellular space, cell corners. Bar = 1 μm.

**Figure 3 cells-10-00685-f003:**
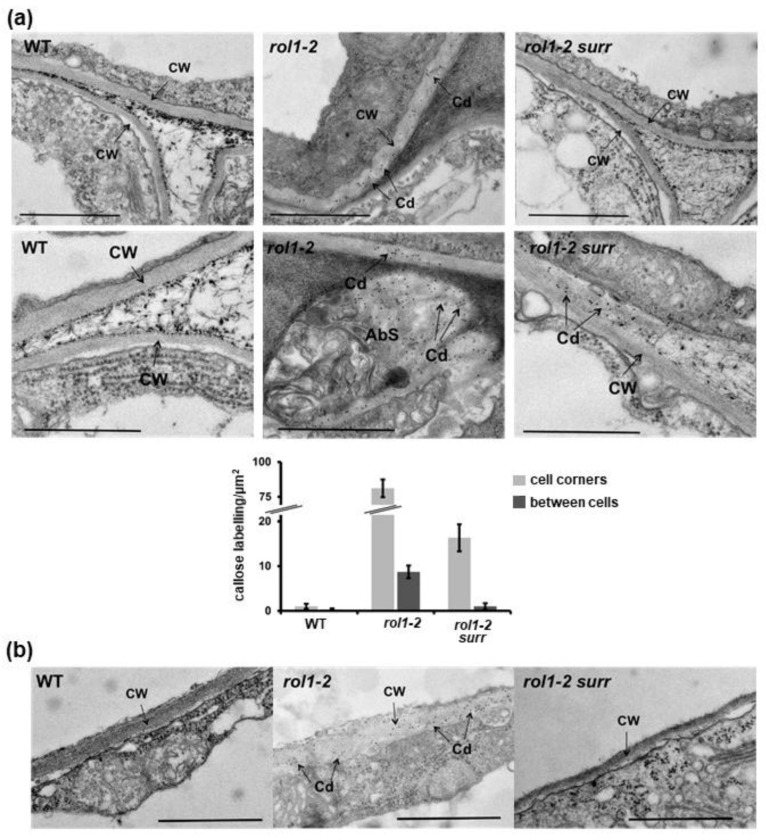
Ectopic callose deposition in cell walls of the *rol1-2* mutant. (**a**) In root tissue, detection of callose revealed no labelling in the wild type (left column, lower picture with stronger magnification), strong labelling in *rol1-2* (middle column) with occurrence of callose deposits (lower picture), and rare labelling in *rol1-2 surr* mutant roots (no and weak labelling in upper and lower picture, respectively). cw: cell wall; cd: callose deposition; AbS: abnormal structures rich in callose. Quantification of callose labelling between the genotypes for cell walls in cell corners and cell walls between neighboring cells. (All differences between genotypes are significant; student *t*-test; *p* < 0.05; *n* > 10). Error bars represent standard error of the mean. Bar = 1 μm.(**b**) In contrast to wild-type root hairs that develop compact cell walls (left), cell walls of *rol1-2* root hairs are rich in callose and have an overall altered, low-electron density appearance (middle). This effect of *rol1-2* is suppressed in the *rol1-2 surr* mutant (right). cw: cell wall; Cd: callose. Bar = 1 μm.

**Figure 4 cells-10-00685-f004:**
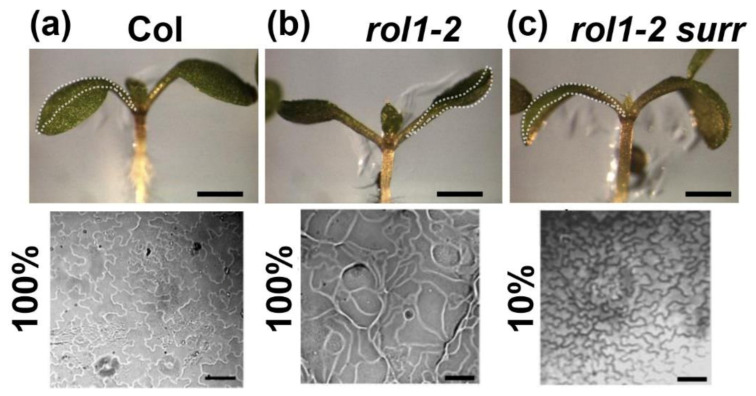
*rol1-2* shoot phenotypes are partially suppressed in *rol1-2 surr*. Seedlings were grown for 6 days and a representative picture of the shoot on side-view was taken (upper row). The bending of the cotyledons is indicated with a stippled line for better visualization. Epinastic cotyledons formed in the wild type (**a**) and the *rol1-2 surr* double mutant (**c**), whereas *rol1-2* formed hyponastic cotyledons (**b**). The lower row shows pavement cell shapes of the adaxial side of cotyledons. Jigsaw puzzle-like cell shapes are typical for the wild type, whereas the lobing is not found in the *rol1-2*. Only 10% of all cotyledons of *rol1-2 surr* mutants show re-establishment of the lobing as shown in (**c**); in 90% of the cases, they develop *rol1-2*-like cell shapes. Barv = 1 mm (upper row), 40 μm (lower row).

**Figure 5 cells-10-00685-f005:**
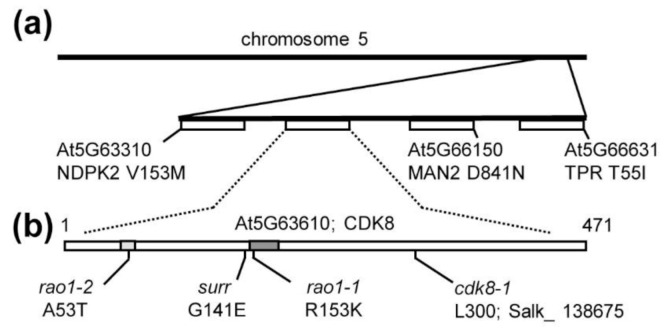
Identification of the *surr* mutation in *cyclin-dependent kinase 8* (*CDK8*). Whole-genome sequencing revealed several genetically linked mutations at the end of chromosome 5—namely, in the genes *nucleotide di-kinase 2* (*NDPK2*), *cyclin-dependent kinase 8* (*CDK8*), *alpha-mannosidase 2* (*MAN2*), and *TPR* (**a**). (**b**) The only mutation completely linked to the *surr* mutant phenotype is in *cycline-dependent kinase 8* (*CDK8*, At5g63610). Previously identified *CDK8* alleles used in this study and the change in amino acids of the missense alleles are indicated. Numbers refer to amino acid positions in the CDK8 protein. The ATP binding site and the kinase active site are indicated by bright and dark grey, respectively.

**Figure 6 cells-10-00685-f006:**
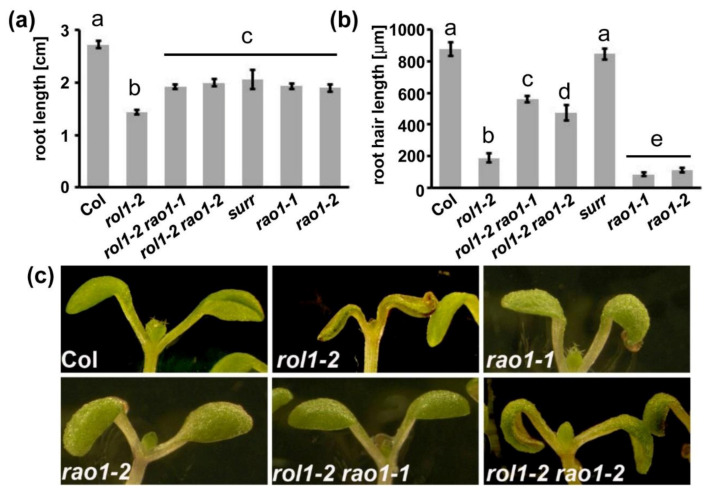
*rao1* alleles suppress the *rol1-2* growth defects. (**a**) Seedlings were grown for 5 days in a vertical orientation and root length was determined. Both *rao1* alleles show shorter roots than the wild type and suppress the *rol1-2* induced short-root phenotype. Significant differences are indicated by letters above the graph (*T*-test, *p* < 0.01, *n* = 20). (**b**) In contrast to *surr*, the *rao1* alleles develop significantly shorter root hairs. In combination with *rol1-2*, however, both *rao1* alleles induce longer root hairs than both *rol1-2* and the *rao1* single mutant. Significant differences are indicated by letters above the graph. (*T*-test, *p* < 0.01, *n* = 50). Error bars represent standard deviations. (**c**) Cotyledon phenotype of the different lines. All except *rol1-2* show epinastic cotyledons. Bar = 1 mm.

**Figure 7 cells-10-00685-f007:**
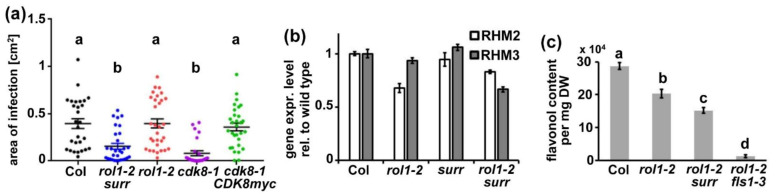
Effect of *surr* on resistance response, gene expression and flavonols. (**a**) Plants were infected with *Botrytis cinerea* and the infected leaf area was quantified. As for the *cdk8-1* T-DNA insertion allele, *surr* also induced an increased resistance whereas *rol1-2* does not affect susceptibility ty. As control, a *cdk8-1* allele transformed with a *CDK8cmyc* construct was included in the experiment. Different letters above the graphs indicate significant differences (Kruskal-Wallis test). (**b**) For quantitative RT-PCR, 7-day-old seedlings were collected and 300 ng of total RNA was used as starting material. Expression levels of *RHM2* and *RHM3* in the wild type were set to 1. (**c**) Flavonols were extracted from shoot tissue of 7-day-old seedling and analyzed by High-performance liquid chromatography (HPLC). The area under each peak was used for quantification. Different letters above the graphs indicate significant differences (*T*-test, *p* < 0.01, *n* = 4). Error bars represent standard errors.

## Data Availability

All material presented here is available upon request: chringli@botinst.uzh.ch. The Appendix A are attached at the end of this file.

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
