# Peer review of "Defects in Cell Wall Differentiation of the Arabidopsis Mutant rol1-2 Is Dependent on Cyclin-Dependent Kinase CDK8"

_cells, 2021, doi:10.3390/cells10030685_

Round 1
Reviewer 1 Report
The manuscript entitled “Defects in cell wall differentiation of the Arabidopsis mutant rol1-2 is dependent on cyclin-dependent kinase” by Schumacher et al. identified a suppressor mutant of rol1-2 named surr. The rol1-2 mutant is a loss-of-function allele of rhamnose synthase 1 (RHM1) and has the defects of root cell elongation and root hair tip growth by aberrant structures and reduction of rhamnogalacturonan (RG) I and II. In this study, the surr mutant is shown to be a weak loss-of-function allele of CDK8. The surr mutation suppressed various developmental defects of rol1-2 including short root hairs, stunted roots (partial recovery), incomplete remodeling of cell wall at the cell corners, callose accumulation, and cotyledon growth, but not epidermal pavement cell morphology and flavonol accumulation. The severe loss-of-function alleles of CDK8 (rao1) also suppressed rol1-2 mutant phenotype. Consequently, CDK8 mediates the defects of cell wall and root development caused by the loss-of-function of RHM1. This study provides novel and original insights on cell wall differentiation during root elongation, in which CDK8 and RHM1 are involved. However, the mode of action of CDK8 remains unknown. Since this manuscript is logical and well written, I have no major concern.
Minor comments
1) I wonder the functional relationships of CDK8 and RHM1 in cell wall differentiation and root growth. Based on protein structure and function, they may function independently. Alternatively, CDK8 may be epistatic: the defects of RHM1 induce CDK8-dependent transcription.
2) I wonder the relationships of CDK8, RHM1, and LRX1 and their relevance to cell wall integrity signaling. LRX1 mediates cell wall integrity signaling through RALF-FER. CDK8 may also mediate transcriptional regulation downstream of cell wall integrity signaling induced by pectin defects in rol1-2. Is this pathway different from RALF-FER-LRX1 signaling or downstream/upstream of RALF-FER-LRX1?
3) Since the double mutants of CDK8 and RHM1 show normal phenotype, both genes are not required for root growth and plant development.
4) What is the phenotype of rol1-2, surr, and rol1-2 surr in lateral roots, inflorescence, flowers, and fertility? Because the rol1-2 surr double mutant is same to the wild type, CDK8 and RHM1 are not required for normal growth and development. The surr single mutants show reduced root growth (Fig. 6). Is there any other phenotype in the surr mutant?
5) The root growth defect of rol1-2 mutant seems to be specific to cell elongation (no defects in root meristem and cell division). The cell wall defect at the cell corners implies its functional importance in cell elongation. It is interesting to examine whether RGI and RGII localize to the cell corners.
Reviewer 2 Report
The manuscript from Isabel Schumacher and colleagues provides new insights regarding the root phenotype of a mutant with altered rhamnogalacturonan I biosynthesis. The authors support the idea that a specific mutation in CDK8 partially alleviates this phenotype through an elegant experimental pipeline. This manuscript is therefore of scientific interest. Experiments were properly conducted and the results are clearly presented and adequately discussed. However, some experiments are lacking to consistently explain the observations raised by the authors and some minor points should be corrected before publication, as further explained below.
- Figure 3: Based on these micrographs, I think it is difficult for the reader to visualise obvious differences in callose deposition between WT and rol1-2, especially as compared to pictures in supplementary data. A quantitative analysis should be performed in the different cell wall regions between the 3 genotypes (densitometric analysis) to convince the reader. In addition, the authors should perform similar analysis targeting the deposition of RG I in the cell wall. Several antibodies such as LM5 (cited by the authors) or INRA-RU1/RU2 are available to detect side chains or backbone of this polysaccharide. Such data would be more convincing than callose deposition and bring new elements concerning the alleviation of rol1-2 phenotype by surr in the root.
- Paragraph 4.1: The authors may perform additional experiments to investigate the lack of cell wall modification in rol1-2, for instance by comparing the evolution of RG-I in the cell corners of the WT between the meristematic zone and the differentiation zone versus rol1-2. The composition of RG-I is known to change between dividing cells and elongating cells (arabinan and galactan side chains for instance), which may be visible by immunocytochemistry (LM5, LM6, INRA RU1-2...). Such experiments will definitely strengthen the manuscript. The authors may also discuss about the possible role of callose in relation with the resistance of mutants to Botrytis cinerea.
- Lines 500-503: among the genes affected by mutations in cdk8, may the authors be more precise about those involved in RG/pectin biosynthesis and modification? This may be of interest for the reader and may help to draw a possible explanation for the compensation mechanism due to the surr mutation.
- Paragraph 2.3: Please specify how you computed the gene expression level indicated in subsequent figures with qPCR results.
- Line 132: please change to aniline
- Paragraph 2.5: Did the authors followed the same fixation and post-fixation protocols for ultrastructural analysis and immunogold labelling? It may be useful to separate the 2 methods.
- Line 175: Accession number of ROL1 seems to be At1g78570
- Line 213: please correct "is remains".
- Figure 3a: The authors may precise they targeted the root differentiation zone in these observations.
- Figure 7b: Albeit the fold-change in expression of RHM2/3 between the WT and rol1-2 / rol1-2 surr mutants is not important (<2), some of them look significant, based on the reported standard deviations. The authors may also merely state that the expression of RHM2 and RHM3 does not compensate the mutation of RHM1.
- Line 404: please change to "Botrytis" and "leaf"
Round 2
Reviewer 2 Report
The authors have improved the manuscript when possible, but some points have not been adressed due to technical reasons. The manuscript is however suitable for publication in Cells.